# Effect of Disulfiram on the Reproductive Capacity of Female Mice

**DOI:** 10.3390/ijms24032371

**Published:** 2023-01-25

**Authors:** Mingming Teng, Yuan Luo, Chan Wang, Anmin Lei

**Affiliations:** Shaanxi Stem Cell Engineering and Technology Research Center, College of Veterinary Medicine, Northwest A&F University, Yangling 712100, China

**Keywords:** disulfiram, female mice, reproductive capacity

## Abstract

In the process of assisted reproduction, the high-oxygen in vitro environment can easily cause oxidative damage to oocytes. Disulfiram (DSF) can play an anti-oxidant or pro-oxidant role in different cells, and the effect of DSF on oocytes remains unclear. Moreover, it remains unclear whether the use of DSF in the early stages of pregnancy has a negative impact on the fetus. In this study, we found that DSF increased serum FSH levels and increased the ovulation rate in mice. Moreover, DSF enhanced the antioxidant capacity of oocytes and contributed to the success rate of in vitro fertilization. Moreover, the use of DSF in early pregnancy in mice increased the uterine horn volume and the degree of vascularization, which contributed to a successful pregnancy. In addition, it was found that DSF regulated the mRNA expression of angiogenesis-related genes (*VEGF*), follicular development-related genes (*C1QTNF3*, *mTOR* and *PI3K*), ovulation-related genes (*MAPK1*, *MAPK3* and *p38 MAPK*) and antioxidant-related genes (*GPX4* and *CAT*). These results indicate that DSF is helpful for increasing the antioxidant capacity of oocytes and the ovulation rate. In early pregnancy in mice, DSF promotes pregnancy by increasing the degree and volume of uterine vascularization.

## 1. Introduction

As the age at which women reproduce increases, ensuring higher-quality oocytes is important [1]. In the process of assisted reproduction, when oocytes leave the in vivo environment, the high-oxygen in vitro environment can easily cause oxidative damage to oocytes [2]; subsequently, oxidative stress reduces the fertilization rate of oocytes [3]. In addition, women use a wide array of high-cost hormone drugs to obtain a sufficient number of transplantable embryos, which can easily result in ovarian hyperstimulation syndrome (OHSS) [4].

Disulfiram (DSF) was officially approved by the Food and Drug Administration (FAD) in 1951 for clinical use as a treatment for alcohol use disorder. Current reports show that disulfiram has the potential to treat other diseases [5,6], such as cancer [7], inflammation [8,9] and obesity [10,11]. The effect of disulfiram on female reproductive capacity remains unclear [12,13]. DSF can induce apoptosis in tumor cells by promoting oxidation [14], increasing the levels of reactive oxygen species and reducing the level of glutathione [15]. However, in the treatment of inflammation, obesity and organ damage, DSF can reduce the intracellular levels of reactive oxygen species [16], increase glutathione level, play an antioxidant role and reduce damage to the body by the disease [10]. It remains unknown whether DSF can promote the oxidation or antioxidation of oocytes.

There is still controversy over whether the use of DSF in early pregnancy affects the fetus. In clinical reports, the use of DSF in the first 3 months of pregnancy was shown to increase the proportion of fetal malformations, and healthy fetuses and continued healthy development were also reported. In addition, women who use DSF to abstain from alcohol may also have drunk alcohol before and during pregnancy, and the adverse effects of alcohol on the fetus are recognized [17]. Therefore, the use of DSF in early pregnancy and its possible effects on offspring deserves further study.

In this study, we evaluated the effect of DSF on the reproductive ability of female mice by detecting the estrous cycle, reproductive hormones, ovulation rate, oocyte antioxidant capacity and in vitro fertilization capacity in mice and examined the effects of DSF in early pregnancy on embryonic development and the number of offspring [18]. Overall, our results showed that 50 mg/kg DSF increased the ovulation rate and the oocyte antioxidant capacity. The use of DSF in early pregnancy in mice increased the uterine horn volume and the degree of vascularization, which contributed to successful pregnancy. In addition, it was found that DSF regulated the mRNA expression of angiogenesis-related genes (VEGF), follicular development-related genes (C1q/tumor necrosis factor-related protein 3, *C1QTNF3*; mammalian target of rapamycin, *mTOR*; Phosphoinositide 3-kinase, *PI3K*) [19], ovulation-related genes (mitogen-activated protein kinase 1, *MAPK1*; mitogen-activated protein kinase 3, *MAPK3* and mitogen-activated protein kinase *14, p38MAPK*) and antioxidant-related genes (glutathione peroxidase 4, *GPX4*; Catalase, *CAT*) [20]. This study provides a theoretical basis for the influence of DSF on female animal reproductive capacity and provides a novel idea for the application of DSF.

## 2. Results

### 2.1. Observation of Vulva

The proestrus of mice was characterized by slightly swollen, pink vulvar folds and a slightly open vaginal orifice (Figure 1A). The estrous period of mice was characterized by crimson and severely swollen vulvar folds, a significantly enlarged vaginal orifice and more viscous secretions (Figure 1B). Vulval swelling in the metestrus of mice gradually subsided, and the color of vaginal mucosa started to become white. A small amount of solidified white secretion was often found (Figure 1C). During diestrus, vulvovaginal swelling subsided, the vaginal opening closed and the vaginal mucosa became pale (Figure 1D).

### 2.2. Vaginal Smears of Mice at Different Stages of the Estrous Cycle

The whole estrous cycle of mice is generally 4–6 days. The vaginal exfoliated cells of mice in the proestrus stage had the largest proportion of nuclear epithelial cells (arrow in Figure 2a), and the overall cell density was low (Figure 2A). The vaginal exfoliated cells in the estrous stage were almost all nuclear-free keratinized epithelial cells (arrow in Figure 2b), and there was generally a high cell density (Figure 2B). The number of leucocytes in vaginal smears in the metestrus stage began to increase, with nuclear-free epithelial cells and leucocytes mainly observed (Figure 2C). The leucocytes (arrow in Figure 2d) accounted for the largest proportion in vaginal exfoliated cells of mice during diestrus (Figure 2D).

### 2.3. Effect of DSF on Estrus Cycle of Mice

Compared with the control group, the duration of diestrus and of the whole estrus cycle of female mice in DSF group was increased (Table 1). After treatment with 50 mg/kg DSF, the duration of diestrus was 2.92 ± 0.28 days, which was longer than that of the control group (1.67 ± 0.22 days, *p* < 0.01). After treatment with 100 mg/kg DSF, the duration of diestrus was prolonged to 3.42 ± 0.15 days (*p* < 0.001). Compared with the control group, the estrous cycle of female mice treated with 50 mg/kg and 100 mg/kg DSF was prolonged from 4.40 ± 0.24 days (control) to 5.60 ± 0.23 days (*p* < 0.01) and 6.25 ± 0.21 days (*p* < 0.001), respectively. The duration of proestrus, estrus and metestrus of female mice after DSF treatment did not significantly change.

### 2.4. Effect of DSF on Reproductive Hormone Levels in Mice

This experiment further explored whether DSF affected the level of reproductive hormones. The mice in diestrus were administered 50 mg/kg DSF i.g., and blood samples were collected from mice in each stage of the estrous cycle. Results showed that, compared with the control group, the serum E2 level in the DSF group was decreased (Figure 3A). Interestingly, the concentration of FSH in serum increased after the use of DSF (Figure 3C). 

### 2.5. Effect of DSF on Ovulation Rate in Mice

It is well established that FSH is essential for follicular development and maturation. Therefore, we studied the effect of DSF on the ovulation rate in mice, and 20 mice were used in each group. Compared with 25.70 ± 1.71% in the control group, the ovulation rate in mice treated with 50 mg/kg DSF reached 37.85 ± 2.35% (Figure 4). Moreover, we found that when the concentration of DSF exceeded 100 mg/kg, the ovulation rate of mice began to decline.

### 2.6. Effect of DSF on Cleavage Rate of Mice Oocytes In Vitro

The cleavage rate is one of the most important indices used to evaluate the quality of oocytes. The results showed that the average cleavage rate of oocytes in mice treated with 50 mg/kg DSF increased from 74.33 ± 1.20% to 80.67 ± 1.21% (Figure 5). To explore the effect of DSF on the antioxidant capacity of oocytes, 50 mg/kg DSF was selected for further study.

### 2.7. Effect of DSF on ROS Level in Mice Oocytes

During the process of in vitro fertilization, oxidative stress is an important factor affecting the development of oocytes [21]; indeed, excessive reactive oxygen species (ROS) inhibit the development of oocytes. The results showed that the level of ROS in mice MII stage oocytes decreased after DSF treatment (Figure 6).

### 2.8. Effect of DSF on GSH Level in Mice Oocytes

Glutathione (GSH) has high importance in maintaining the quality of oocytes; it is an essential antioxidant in oocytes. A decrease in GSH leads to DNA damage and increased levels of ROS in MII stage oocytes. The relative fluorescence intensity results showed that GSH level in mice MII stage oocytes was higher after DSF treatment (Figure 7). The above results indicated that DSF decreased the ROS levels in oocytes, increased the GSH level and enhanced the antioxidant capacity of oocytes; these changes are beneficial to the further development of oocytes.

### 2.9. Effect of DSF on Mice Embryo Development

The morning after the female mice were mated, the embolus was checked. The mice were administered 50 mg/kg DSF once per day for 3 days, and the blastocysts in the mouse uterine horn were collected on the fourth night. The number of female mouse blastocysts increased after DSF treatment. The average number of blastocysts per mouse was 14.67 ± 1.69 and 23.33 ± 1.56 in the control and DSF groups, respectively (Figure 8).

### 2.10. The Influence of DSF on the Uterus in Pregnant Mice 

During the collection of mouse blastocysts, the degree of vascularization of the uterine horn of mice was increased after DSF treatment, with more capillaries and blood clearly present on the surface of uterine horn. In addition, the volume of the uterine horn was larger in the DSF group. Its diameter increased from 2.00 ± 0.13 mm in the control group to 3.17 ± 0.28 mm in the DSF group (Figure 9) with no significant difference in the length. 

### 2.11. Influence of DSF on Mouse Offspring

DSF was used for three consecutive days after the female mice were mated, and the conception rate, litter size and offspring weight were measured. The conception rate in female mice in the DSF group was increased by 22% compared with the control group (Table 2). There was no significant difference in the average litter size and birth weight between the control and DSF groups.

### 2.12. Effect of DSF on Expression of Angiogenesis-Related Genes

To further explore the mechanism through which DSF increases the degree of vascularization, the mRNA expression of *VEGF* was quantitatively detected. VEGF is an important growth factor that has angiogenic activity. The high level of VEGF secreted by granulosa cells in follicles is positively correlated with a higher fertilization rate of oocytes, better embryo quality and higher pregnancy rate [22]. The mouse granulosa cells were treated with different concentrations of DSF for 24 h, and then the fluorescence quantitative detection was performed. DSF concentrations of 0.25 μM, 0.5 μM and 1 μM increased the relative expression of *VEGF* mRNA (Figure 10). This result indicates that DSF may promote angiogenesis by upregulating the expression of *VEGF*, consequently improving the fertilization rate of oocytes, embryo quality and pregnancy rate.

### 2.13. Effect of DSF on Expression of Follicular Development-Related Genes

To further explore the mechanism through which DSF increases ovulation in mice, the expression of genes related to follicular development and maturation was detected (C1QTNF3, mTOR and PI3K). Results showed that 0.25 μM, 0.5 μM and 1 μM DSF upregulated the expression of genes related to follicular development and maturation (Figure 11). It is suggested that DSF may promote the development and maturation of follicles by upregulating the expression of C1QTNF3, mTOR and PI3K.

### 2.14. Effect of DSF on Expression of Ovulation-Related Genes

The effects of different concentrations of DSF on genes related to follicular development and maturation were detected, along with the expression of ovulation-related genes (*MAPK1*, *MAPK3* and *p38MAPK*). Results showed that 0.25 μM, 0.5 μM and 1 μM DSF upregulated the expression of ovulation-related genes (Figure 12).

### 2.15. Effect of DSF on Expression of Antioxidant-Related Genes

To explore the mechanism through which DSF reduces ROS levels and increases the GSH level in oocytes, which improves the ability of oocytes to resist oxidative stress, the primary granulosa cells of mice were cultured, and DSF (0.25 μM, 0.5 μM, 1 μM and 10 μM) was added to the culture medium. After the samples were collected, the relative mRNA expression of the antioxidant-related genes *CAT* and *GPX4* was detected. Results showed that 0.25 μM, 0.5 μM and 1 μM DSF promoted the expression of *CAT* and that 0.5 μM and 1 μM DSF promoted the expression of *GPX4* (Figure 13). These results indicated that DSF might enhance the antioxidant capacity of oocytes by upregulating the expression of *CAT* and *GPX4*.

## 3. Discussion

As a drug for the treatment of alcohol use disorder, DSF has been used clinically for nearly 70 years; it has a low price and few adverse reactions are encountered [23]. Assisted reproduction has come to play an increasingly important role in the field of fertility. The quantity and quality of ovulation greatly affect the success rate of assisted reproduction. However, the large number of hormone drugs required can easily lead to female OHSS and is expensive. In recent years, it has been reported that DSF can play an important role in the treatment of cancer [24], obesity [25], inflammation [26,27], sepsis [28] and arrhythmia [29], but the effect of DSF in female reproduction is less studied [30]. Based on the single maximum safe dose in humans, the mouse dose was calculated as 88 mg/kg [31]. In a study on squamous cell carcinoma, a DSF concentration of 50 mg/kg significantly inhibited tumor growth, and the experimental animals maintained normal metabolism and stable weight during the experiment [32]. DSF can prevent weight gain in mice [25]. After ingestion of DSF, the energy consumption in mice increased, the circulating leptin concentration decreased [33] and the fat mass decreased, but the food intake was significantly changed [10]. DSF can prevent and treat various inflammatory conditions and organ injuries by inhibiting oxidative stress and the activation of the NOD-like receptor thermal protein domain associated protein 3 (NLRP3) inflammasome and further inhibiting cell scorch [34,35].

In this study, we assessed the estrous cycle, hormone level, ovulation rate, oocyte quality, embryonic development, pregnancy rate and litter size to determine whether DSF affected the reproductive capacity of female mice. We found that the use of DSF during diestrus could increase the duration of diestrus, thus increasing the time for the whole estrus cycle. The serum was then collected and tested, and it was found that DSF increased FSH level. FSH is essential for oocyte development and maturation, so we examined the ovulation rate in mice. The results showed that 50 mg/kg DSF increased the ovulation rate in mice (*p* < 0.001). Consequently, we explored the effect of DSF on the quality of oocytes further. The cleavage rate of oocytes in vitro is one of the most important indices used to evaluate oocyte quality. Results showed that the average cleavage rate of oocytes was higher in the 50 mg/kg DSF group than in the control group. Moreover, we observed that when the concentration of DSF exceeded 100 mg/kg, the ovulation rate and cleavage rate of fertilized oocytes in vitro began to decline. Although there was no statistically significant difference, the use of DSF exceeding the maximum safe dosage may have a negative effect on female reproductive capacity.

Under normal physiological conditions, the level of ROS in oocytes is maintained within a relatively low range [36]. The antioxidant capacity of oocytes is in a dynamic equilibrium with the levels of ROS, and ROS can participate in the maturation process of oocytes [37]. However, when oocytes are exposed to the high-oxygen environment in vitro, this dynamic balance can be disrupted [38]. Excessive ROS content in oocytes will induce oxidative stress, damage the mitochondria, reduce mitochondrial membrane potential, reduce oocyte quality and affect further embryonic development [39]. GSH is the main non-enzymatic antioxidant against ROS in oocytes and embryos, and the decrease in GSH level is the signal for cells to begin apoptosis [40]. GSH in oocytes can protect spindles and proteins from oxidative damage, maintain normal spindle morphology and function and ensure correct meiosis, which is necessary for zygote formation after fertilization [41]. The increase in GSH level in oocytes helps to promote oocyte cytoplasmic maturation, improve oocyte quality and protect oocytes from oxidative damage during embryonic development after fertilization [42]. In addition, there are other enzymes in oocytes, such as CAT and superoxide dismutase (SOD), and their concentration also affects the quality of oocytes.

The regulatory effect of DSF on oxidative stress is cell-specific. In tumor cells, DSF can promote tumor cell apoptosis by decreasing the GSH level and mitochondrial membrane potential but increasing intracellular ROS levels [43]. In LPS-induced inflammation and organ damage, DSF can inhibit oxidative stress by increasing GSH and SOD levels and decreasing Malonydialdehyde (MDA) levels [35,44]. The antioxidant capacity of oocytes is an important factor affecting the quality of oocytes [45]. The overproduction of ROS can lead to oxidative stress, which reduces the quality of oocytes and their development potential [46]. In this experiment, 50 mg/kg DSF reduced ROS levels in oocytes (*p* < 0.01). GSH is an important antioxidant molecule. Once the oocyte leaves the internal environment, it experiences hyperoxic stress, so GSH is required to resist the damage caused by oxidation. Compared with the control group, the GSH level was significantly higher in the 50 mg/kg DSF group (*p* < 0.01). The above results indicate that DSF may improve the quality of oocytes by increasing the GSH level and decreasing ROS levels in oocytes, which enhances the ability of oocytes to resist oxidative stress.

It remains unclear whether the fetal malformation is caused by alcohol or DSF itself [47]. In this study, it was found that DSF affected mouse embryo development, with the average number of blastocysts per mouse increased by DSF treatment. In addition, it was also observed that the degree of uterine vascularization was higher and that the uterine volume was larger in mice after DSF treatment. The increase of uterine vascularization can provide sufficient blood circulation for the fetus, and the hormones and nutrients brought by the blood contribute to embryo development, implantation and further development [48]. Interestingly, the pregnancy rate in rats was increased by 22% after DSF treatment, with no significant difference in the average litter size between the DSF and control groups (Table 2). This indicated that DSF treatment in early pregnancy had no obvious negative effect on the offspring.

Whether for the growth and development of follicles, the development and implantation of embryos, or the growth and development of embryos after implantation, sufficient blood supply is needed to provide nutrients, oxygen and hormones [49]. The inhibition of angiogenesis can lead to follicular development, ovulation and pregnancy interruption [50]. The VEGF has a vital role in angiogenesis. In follicles, VEGF is mainly secreted by granulosa cells [51], and the high concentration of VEGF is positively correlated with the increased density in blood vessels around follicles [52], a high fertilization rate, good embryo quality and a high pregnancy rate [22]. The upregulation of VEGF helps to increase endometrial vascular permeability and stimulate endometrial angiogenesis and enables the embryo to directly induce angiogenesis at the implantation site, which is essential to improve the implantation rate of the embryo [48]. Clinically, the expression of VEGF in the endometrium of infertile patients and patients with repeated spontaneous abortion is low [53]. The primary mouse ovarian granulosa cells were isolated and cultured, and then different concentrations of DSF (0.25 μM, 0.5 μM, 1 μM, 10 μM and 40 μM) were added to the cell culture solution. After 24 h, cell growth was observed. The results showed that 40 μM DSF caused significant inhibition of the proliferation of mouse granulosa cells, resulting in abnormal cell morphology. However, the proliferation ability and morphology of mouse granulosa cells were not affected by 0.25 μM, 0.5 μM, 1 μM and 10 μM DSF treatment, so we performed a quantitative analysis of these cells. The results showed that the expression of *VEGF* was upregulated by DSF within a certain concentration range. Combined with the results of in vivo experiments, we speculated that DSF may increase the degree of vascularization by upregulating VEGF and promoting conception in female mice. In vivo experiments in mice showed that 50 mg/kg DSF significantly increased the ovulation rate in mice and improved the antioxidant capacity of oocytes, so the relative mRNA expression of follicular development and maturation-, ovulation- and antioxidant-related genes was detected. C1QTNF3 was mainly detected in granulosa cells and in the oocytes of dominant follicles. The mRNA expression of C1QTNF3 was upregulated by FSH [54]. The absence of C1QTNF3 increased the number of atretic follicles and impaired their maturation [55]. The PI3K pathway is the core signaling pathway controlling primordial follicle activation [56]. mTOR is important for maintaining follicular development [57], the fecundity of female mice, the integrity of the oocyte genome, oocyte quality and the growth and differentiation of granulosa cells [58,59]. The PI3K/AKT/mTOR signaling pathway is also involved in angiogenesis [60]; p38MAPK plays an important role in follicular development, oocyte maturation and ovulation [61,62]. The inhibition of p38MAPK expression will hinder oocyte meiotic recovery and cumulus expansion [63,64]. MAPK3/1 is essential for the recovery of meiosis [65], ovulation and luteinization in oocytes stimulated by LH [66,67]. The results showed that DSF upregulated the relative mRNA expression of genes related to follicular development and maturation (*C1QTNF3*, *mTOR* and *PI3K*) and ovulation (*MAPK1*, *MAPK3* and *p38MAPK*). Moreover, DSF promoted the expression of antioxidant-related genes (*GPX4* and *CAT*).

## 4. Materials and Methods

### 4.1. Ethics Statement

All animal procedures and study design were conducted in accordance with the Guide for the Care and Use of Laboratory Animals (Ministry of Science and Technology of China, 2006) and approved by the Animal Ethics Committee of Northwest A&F University. ICR female mice (8–10 weeks old) and ICR male mice (10–12 weeks old) used for this experiment were purchased from the Chengdu Dossy Experimental Animals Co. Ltd. (certificate no.: SCXK [CHUAN] 2020–030). The mice were maintained in the animal facility of the Laboratory of Veterinary Medicine, Northwest A&F University under a 12-h light/dark cycle with free access to food and water.

### 4.2. Reagents

Unless otherwise stated, all chemicals were purchased from Sigma-Aldrich Chemie (Schnelldorf, Germany). Disulfiram was purchased via MCE (Shanghai, China). Dulbecco’s Modified Eagle Medium/Nutrient Mixture F-12 (DMEM/F-12) and phosphate-buffered saline (PBS) were purchased via Gibco (Shanghai, China). Fetal bovine serum was purchased via HyClone (Shanghai, China). Pregnant mare serum gonadotropin (PMSG) and human chorionic gonadotropin were purchased from Sansheng (Ningbo, China). The Reactive Oxygen Species Assay Kit was purchased via Beyotime (Shanghai, China). The Glutathione Assay Kit was purchased via Thermo Fisher Scientific (Shanghai, China). Kits for total RNA extraction, reverse transcription and fluorescence quantitation kit were purchased via Tiangen Biotech (Beijing, China). The HE Staining Kit was purchased via Solarbio (Beijing, China). The primer synthesis was conducted by Tsingke Biotech (Beijing, China). The sequences were searched in the NCBI BLAST database.

### 4.3. DSF Preparation

The DSF dose was calculated based on mouse body weight. An appropriate amount of DSF was weighed. For the example of a 1 mL working solution, 0.1 mL of DMSO and 0.9 mL of PEG300 were added in turn, and an ultrasound was used to aid dissolution. The control group received the same amount of solvent without DSF. For each mouse, a single dose was not permitted to exceed 0.2 mL.

### 4.4. Observation of Vulva and Vaginal Smear

By observing the color, degree of swelling and vaginal orifice of the vulva mucosa, a preliminary determination of the estrus period in mice was made, which was then further confirmed by a vaginal smear. Physiological saline was taken up with a 10 μL pipette; the tip of the pipette was gently inserted into the vaginal opening for 1–2 mm and rinsed repeatedly 2 or 3 times, and then a small drop of sample was placed evenly on the slide and allowed to dry. Thereafter, 95% ethanol was dropped on sample area to fix it. After drying, the sample was stained with hematoxylin dye solution for 10 min and gently rinsed off with distilled water, and then the sample was stained with eosin dye solution for 1 min, gently rinsed off with distilled water, dried and observed under a microscope. The dyeing time was adjusted according to the dyeing results. In this experiment, to explore the effect of DSF on the estrous cycle in mice, female mice in diestrus were selected. The drug group was treated with DSF i.g. (DSF group, *n* = 10), whereas the control group (*n* = 10) was treated with an equal amount of solvent. From the second day after administration, the estrus period was identified and recorded at 08:00 and 20:00 every day.

### 4.5. Blood Collection and Hormone Detection

Female mice in diestrus were selected. The drug group (*n* = 12) was treated with 50 mg/kg DSF i.g. while the control group (*n* = 12) was treated with an equal amount of solvent. From the second day after administration, the changes in the mice’s estrus period were observed and recorded, and blood samples from two groups of mice were collected at proestrus, estrus, metestrus and diestrus. Female mice were anesthetized with an intraperitoneal injection of 0.1 mL/10 g of 4% chloral hydrate. After 5 min, the mice were anesthetized, and blood was collected from the eyeballs. The blood samples were allowed to stand for 0.5–1 h and then centrifuged at 500× *g* for 10 min. The upper layer (serum) was transferred to a new centrifuge tube, and the serum hormone content was detected by ELISA (Enzyme-linked Immunosorbent Assay) in strict accordance with the kit instructions.

### 4.6. Collection of Oocytes

Female mice were intraperitoneally injected with 10 IU PMSG (Pregnant Mare Serum Gonadotropin). After 42 h, the experimental group was intraperitoneally injected with different concentrations of DSF i.p., whereas the control group received an equal amount of solvent. After 6 h, 10 IU hCG (human Chorionic Gonadotropin) was intraperitoneally injected. At 16 h after hCG injection, the mice were euthanized by cervical dislocation. The fallopian tubes of female mice were collected and placed in preheated PBS solution. The swollen part of fallopian tube was torn under a microscope, the cumulus cells were removed with 0.25% hyaluronidase, and the treatment was stopped immediately when the cumulus layer began to spread. 

### 4.7. In Vitro Fertilization (IVF)

The epididymis was collected from male mice, and the epididymis tail was punctured. The sperm were then squeezed into the capacitive solution (HTF). The sperm were incubated at 37 °C and 5% CO_2_ for 30–60 min to capacitate the sperm, and then suspension fluid was added to obtain a sperm concentration of 1 × 10^6^/mL sperm. Mature oocytes were washed twice in HTF and fertilized in the medium containing fresh sperm for 5 h. The formation of two pronuclei was regarded as a sign of successful fertilization. The oosperm were washed three times in KSOM and then cultured in KSOM in 95% humidified air with 5% CO_2_ at 37 °C. The cleavage rate was calculated on the second day after IVF. IVF was repeated 3 times; at least 100 oocytes were treated per group.

### 4.8. Reactive Oxygen Species (ROS) Detection

The level of ROS in mouse oocytes was detected with a DCFH-DA reagent. The oocytes were transferred to the preheated DCFH-DA reagent diluted with M2 medium to yield a final DCFH-DA concentration of 10 μm. The oocytes were incubated at 37 °C for 20 min and then washed three times in M2 solution. After washing, they were visualized using a fluorescence microscope (Nikon, Tokyo, Japan). ImageJ software was used to analyze the fluorescence intensity. ROS staining was repeated 3 times; at least 30 oocytes were treated per group.

### 4.9. L-Glutathione (GSH) Detection

Oocytes or embryos were collected, transferred into 20 μM ThiolTracker and stained at 37 °C for 30 min. After staining, they were washed 3 times with PBS, observed, and photographed using a fluorescence microscope (Nikon, Tokyo, Japan). The fluorescence intensity was analyzed with ImageJ software. GSH staining was repeated 3 times; at least 30 oocytes were treated per group.

### 4.10. Collection of Blastocysts

Female mice were intraperitoneally injected with 10 IU PMSG and then with 10 IU hCG 48 h later. Thereafter, they were mated with adult fertile male mice. At 08:00 the next morning, the vaginal orifice of the female mice was checked. If vaginal plugs were observed, the mating was considered successful. The successfully mated female mice were gavaged with 50 mg/kg DSF i.g. once a day for 3 consecutive days. On the fourth night, the mice were euthanized by cervical dislocation, the uterine horns on both sides were collected, and the joint between uterine horns and fallopian tubes was cut longitudinally. At the incision position, a 1 mL syringe was inserted containing flushing fluid, and the uterine cavity was flushed toward the cervix. The samples were separated into a control group (*n* = 6) and DSF group (*n* = 6).

### 4.11. Determination of Pregnancy Rate and Litter Size

After successful mating, female mice were administered 50 mg/kg DSF i.g. once a day for 3 days. The pregnant mice produced offspring in approximately 21 days, and the litter size and offspring weight were recorded. The samples were separated into a control group (*n* = 20) and DSF group (*n* = 18).

### 4.12. Isolation and Culture of Ovarian Granulosa Cells

Female mice were euthanized by cervical dislocation. The ovaries of mice were separated with sterile instruments and washed three times with PBS containing antibiotics, and the surrounding fat and capsule were removed under a microscope. After washing, the ovaries were transferred to DMEM-F12 medium containing antibiotics, and the follicles were punctured with a 1 mL syringe needle under a microscope. The granular cells were released from the ruptured follicles into the culture medium, transferred to a centrifuge tube, and centrifuged at 800 r/min for 5 min, after which the supernatant was discarded. The cells were resuspended in DMEM-F12 culture solution containing antibiotics, mixed evenly, counted and inoculated in a cell culture dish. The cells were cultured in a 5% CO_2_ incubator at 37 °C for 48 h, and then the solution was changed to remove the unattached cells. The culture of granular cells was continued by adding cell culture medium containing different concentrations of DSF (0.25 μM, 0.5 μM, 1 μM or 10 μm; ensuring the DSF in the final working solution did not exceed 0.01% of the volume of the culture solution), and the control group were grown in a DMEM-F12 culture solution without DSF.

### 4.13. RNA Extraction and Quantitative Reverse Transcription PCR (RT-qPCR)

To explore the possible mechanism through which DSF regulates the reproductive capacity of female mice, ovarian granulosa cells were cultured in cell culture medium containing different concentrations of DSF (0, 0.25 μM, 0.5 μM, 1 μM or 10 μM). After culturing for 24 h, a total RNA extraction kit was used to extract RNA from granular cells in different concentration groups, and the concentration and purity of RNA were determined to be within the available range. The number of granular cells in each sample was about 2 × 10^6^. A reverse transcription kit was used to reverse-transcribe RNA into cDNA for real-time fluorescence quantitative assay. The relative mRNA expression of each gene was detected with qRT-PCR. The primer sequences of genes were designed using Primer-BLAST (https://www.ncbi.nlm.nih.gov/tools/primer-blast/index.cgi?LINK_LOC=BlastHome) (accessed on 17 October 2020), with β-actin used as a housekeeping gene. The mRNA expression of related genes was detected using SuperReal PreMix Color (Tiangen Biotech, Beijing, China) and qRT-PCR conditions of 95 °C for 15 min followed by 40 cycles of 95 °C for 10 s and 60 °C for 32 s). The detection results were analyzed with the Bio-Rad CFX Maestro program.

### 4.14. Statistical Analysis

The GraphPad Prism software was used to perform all statistical analyses. One-way analysis of variance with Tukey’s multiple comparison test was performed, and *p*-values less than 0.05 were considered significant.

## 5. Conclusions

In conclusion, DSF increased the concentration of FSH, increased the ovulation rate of mice and enhanced the antioxidant capacity of oocytes. In addition, DSF promoted conception in mice by increasing the degree of vascularization. Our study has found the potential application value of DSF in improving the reproductive capacity of female animals, extended the theoretical basis for the function of DSF in the field of female reproduction and provided new ideas for improving assisted reproduction.

## Figures and Tables

**Figure 1 ijms-24-02371-f001:**
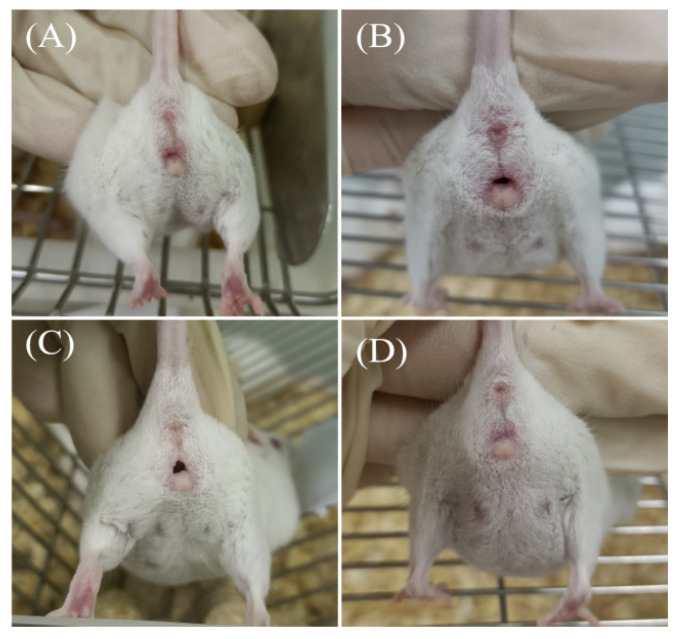
Vulvovaginal characteristics of mice. (**A**) Proestrus. Slightly swollen pink vulvar folds and slightly open vaginal orifice. (**B**) Estrus. Crimson and severely swollen vulvar folds, significantly enlarged vaginal orifice and more viscous secretions. (**C**) Metestrus. The vulvar swelling gradually subsided, and the color of the vaginal mucosa became white. (**D**) Diestrus. Vulvovaginal swelling subsided, vaginal openings closed and pale vaginal mucosa.

**Figure 2 ijms-24-02371-f002:**
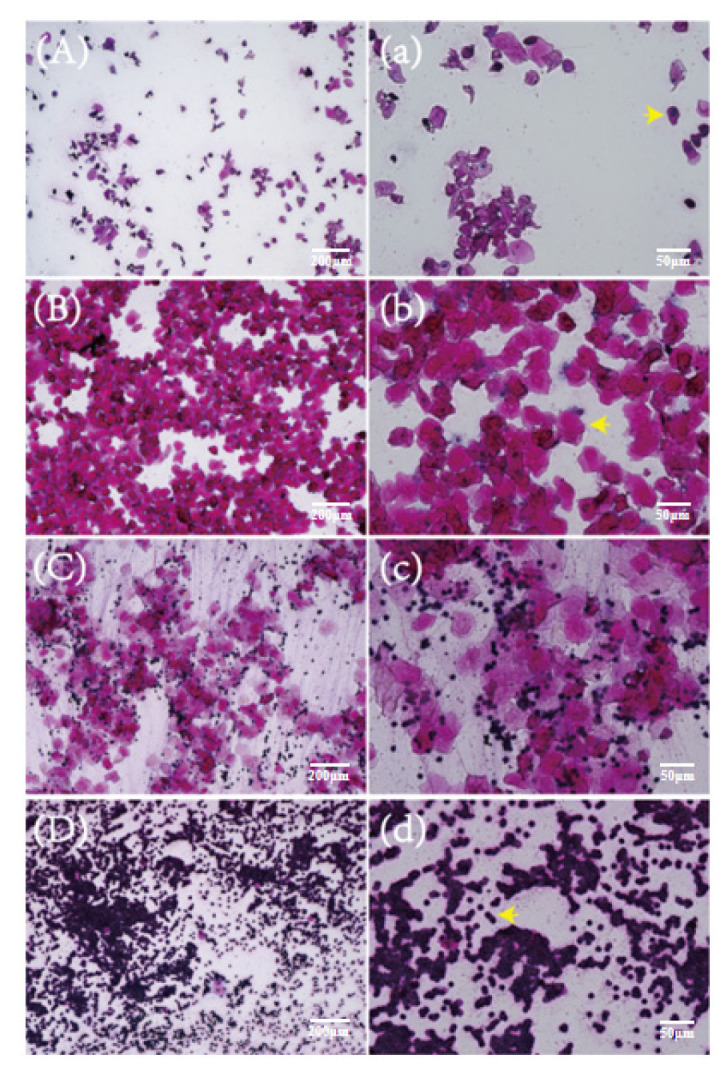
Vaginal exfoliated cells in different stages of the estrus cycle. (**A**) Proestrus. Nucleated epithelial cells account for the largest proportion. (**B**) Estrus. Nuclear-free keratinized epithelial cells account for the largest proportion. (**C**) Metestrus. The number of leukocytes began to increase, and the cells were mainly denucleated keratinocytes and leukocytes. (**D**) Diestrus. Leukocytes account for the largest proportion. (**A**–**D**) scale bar = 200 μm. (**a**–**d**) scale bar = 50 μm. The arrow in (**a**) indicates nucleated epithelial cells; the arrow in (**b**) indicates nuclear-free keratinized epithelial cells; the arrow in (**d**) indicates leukocytes.

**Figure 3 ijms-24-02371-f003:**
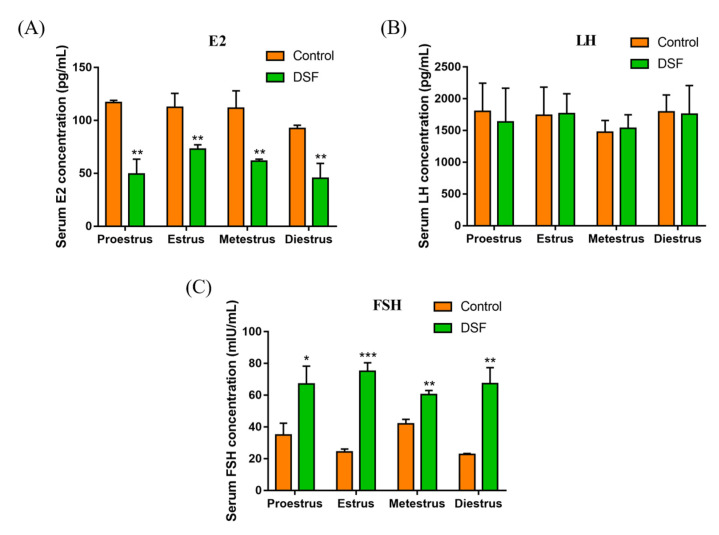
Effect of DSF on reproductive hormones in female mice. (**A**) Concentrations of E2 in serum of control group and DSF group mice during the estrous cycle; (**B**) concentrations of LH in serum of control group and DSF group mice during the estrous cycle; (**C**) concentrations of FSH in control group and DSF group mice during the estrous cycle; * *p* < 0.05, ** *p* < 0.01, *** *p* < 0.001. Control group, *n* = 12; 50 mg/kg DSF group, *n* = 12.

**Figure 4 ijms-24-02371-f004:**
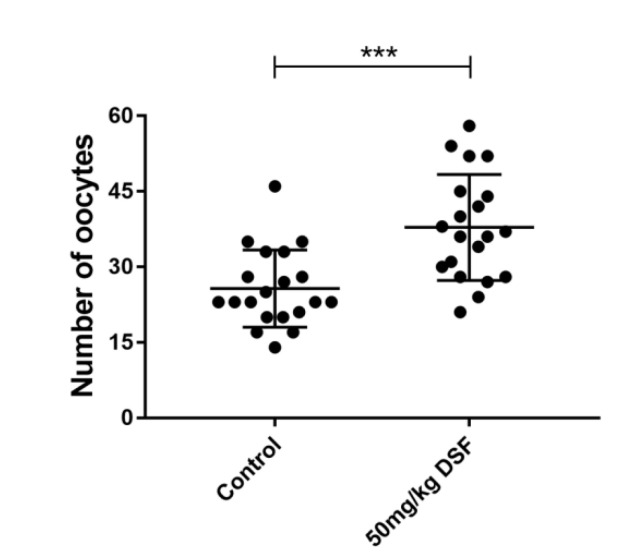
Effects of different concentrations of DSF on ovulation rate in mice. *** *p* < 0.001; Control group, *n* = 20; 50 mg/kg DSF group, *n* = 20.

**Figure 5 ijms-24-02371-f005:**
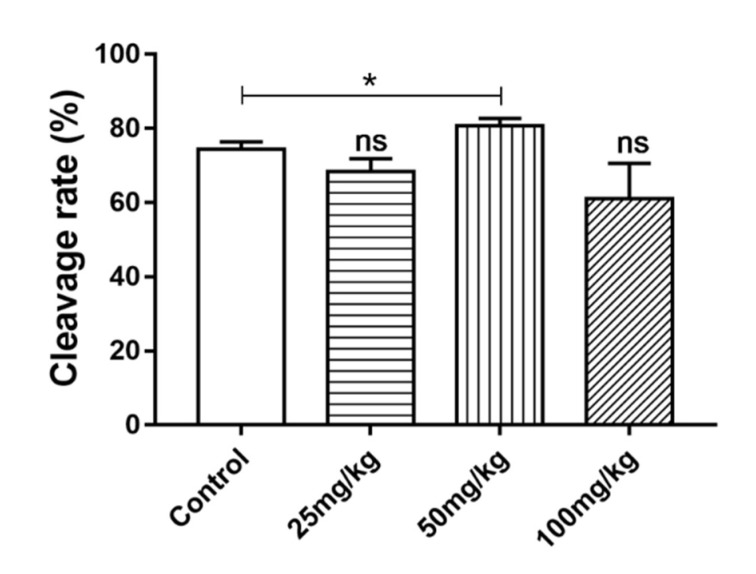
Effects of different concentrations of DSF on cleavage rate of mice oocytes in Vitro. * *p* < 0.05, ns *p* > 0.05. Each group tested more than 100 MII oocytes.

**Figure 6 ijms-24-02371-f006:**
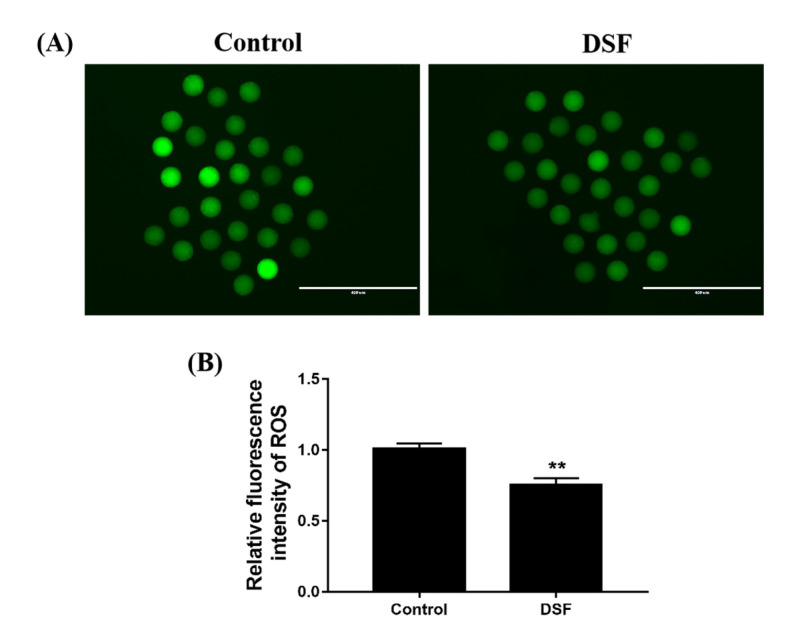
Effect of DSF on ROS level of mouse oocytes. (**A**) Representative images of ROS staining in the control and DSF groups; (**B**) the relative fluorescence intensity of ROS in the control and DSF groups. ** *p* < 0.01; scale bar = 400 μm. Each test evaluated at least 30 MII oocytes in each group, and repeated 3 times.

**Figure 7 ijms-24-02371-f007:**
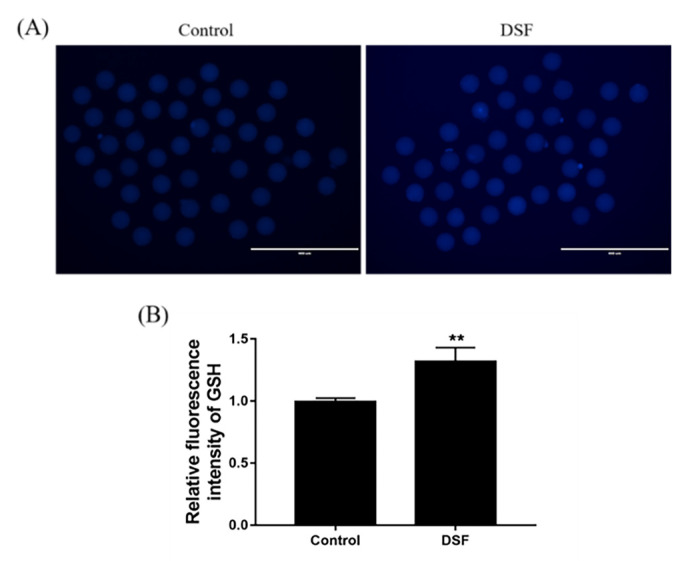
Effect of DSF on GSH level in mouse oocytes. (**A**) Representative images of GSH staining in the control and DSF groups; (**B**) relative fluorescence intensity of GSH in the control and DSF groups. ** *p* < 0.01; scale bar = 400 μm. Each test evaluated at least 30 MII oocytes in each group, and repeated 3 times.

**Figure 8 ijms-24-02371-f008:**
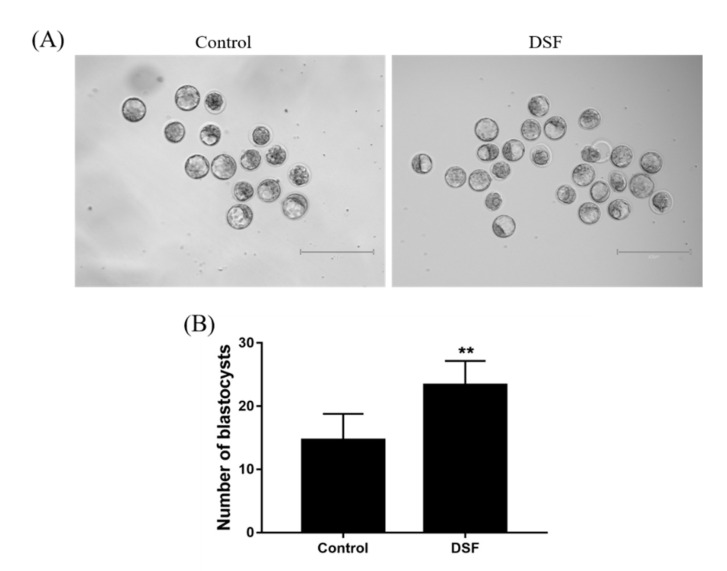
Effect of DSF on the number of blastocysts in mice. (**A**) Representative images of the number of blastocysts per mouse in the Control and DSF groups; (**B**) Comparison of the number of blastocysts per mouse in the Control and DSF groups; ** *p* < 0.01; scale bar = 400 μm. Control group, *n* = 6; 50 mg/kg DSF group, *n* = 6.

**Figure 9 ijms-24-02371-f009:**
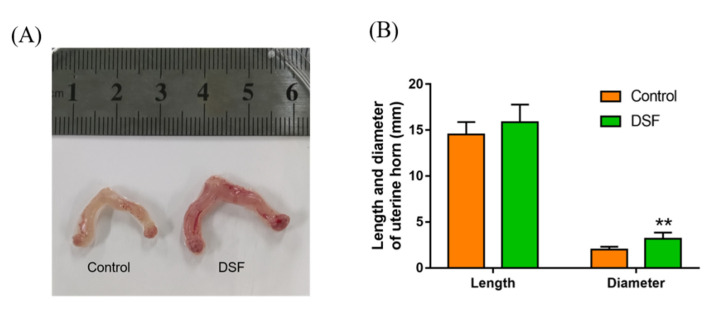
Effect of DSF on degree of vascularization and the length and diameter of the uterine horn in pregnant mice. (**A**) Representative images of uterine horn vascularization in the control and DSF groups; (**B**) the length and diameter of the uterine horn in the control and DSF groups. ** *p* < 0.01. Control group, *n* = 8; 50 mg/kg DSF group, *n* = 8.

**Figure 10 ijms-24-02371-f010:**
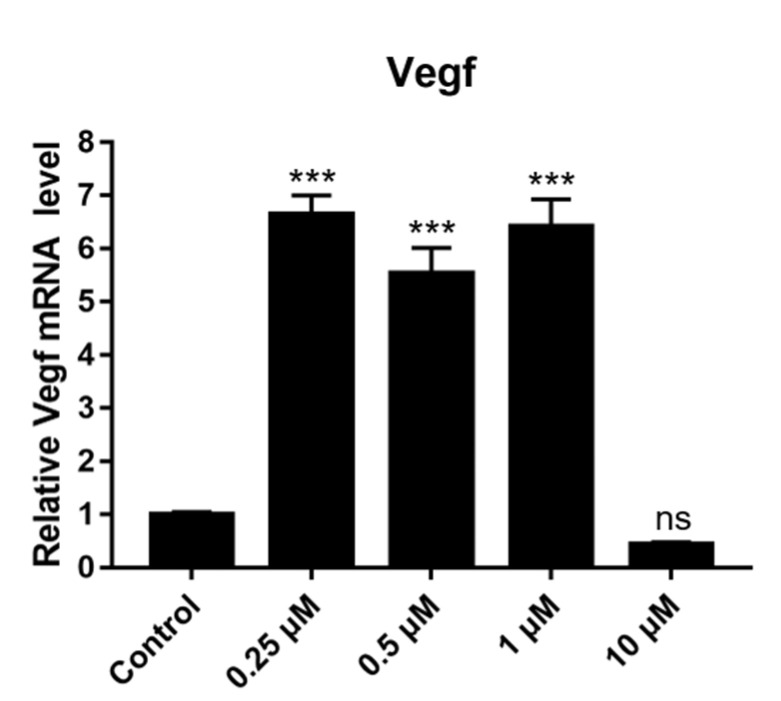
Effects of different concentrations of DSF on the relative mRNA expression of *VEGF*. *** *p* < 0.001, ns *p* > 0.05.

**Figure 11 ijms-24-02371-f011:**
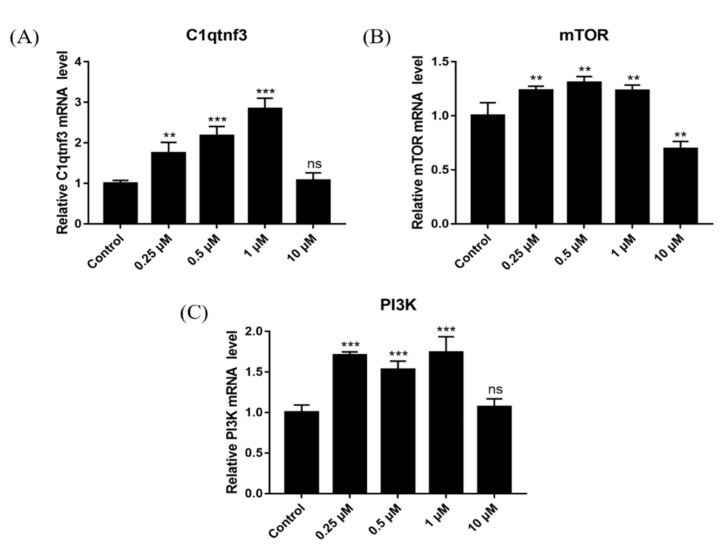
Effects of different concentrations of DSF on the relative mRNA expression of genes related to follicular development and maturationm. (**A**) relative expression of *C1qtnf3* mRNA in different groups; (**B**) relative expression of *mTOR* mRNA in different groups; (**C**) relative expression of *PI3K* mRNA in different groups. ** *p* < 0.01, *** *p* < 0.001, ns *p* > 0.05.

**Figure 12 ijms-24-02371-f012:**
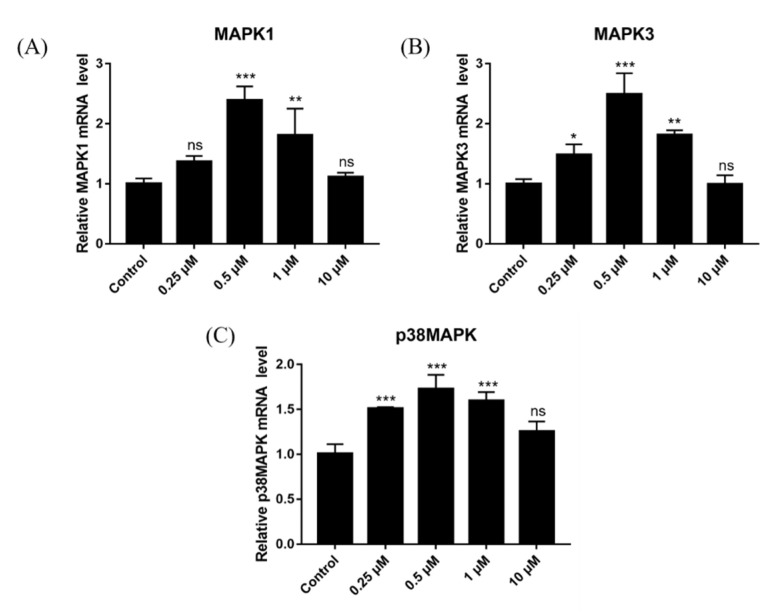
Effects of different concentrations of DSF on the relative mRNA expression of ovulation-related genes. (**A**) relative expression of *MAPK1* mRNA in different groups; (**B**) relative expression of *MAPK3* mRNA in different groups; (**C**) relative expression of *p38MAPK* mRNA in different groups. * *p* < 0.05, ** *p* < 0.01, *** *p* < 0.001, ns *p* > 0.05.

**Figure 13 ijms-24-02371-f013:**
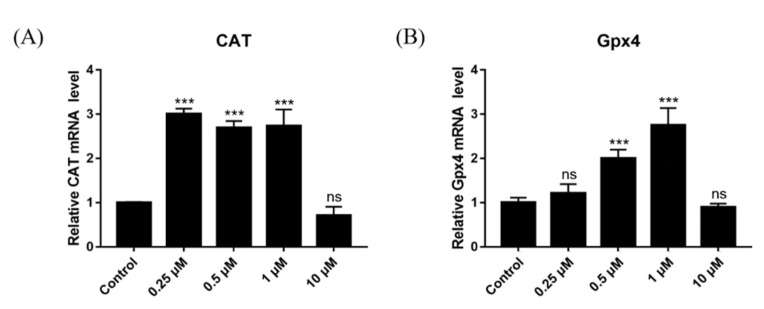
Effects of different concentrations of DSF on the relative mRNA expression of antioxidant-related genes. (**A**) relative expression of *CAT* mRNA in different groups; (**B**) relative expression of *Gpx4* mRNA in different groups. *** *p* < 0.001, ns *p* > 0.05.

**Table 1 ijms-24-02371-t001:** Effects of different concentrations of DSF on the estrous cycle in mice.

Duration of Different Estrus Periods (Days)	Control	DSF 50 mg/kg	DSF 100 mg/kg
Proestrus	0.75 ± 0.11	1.00 ± 0.16	1.00 ± 0.13
Estrus	1.17 ± 0.17	1.00 ± 0.13	0.92 ± 0.08
Metoestrus	0.83 ± 0.11	0.67 ± 0.11	0.92 ± 0.15
Dioestrus	1.67 ± 0.22 ^a^	2.92 ± 0.28 ^b^	3.42 ± 0.15 ^b^
Estrus cycle	4.40 ± 0.24 ^a^	5.60 ± 0.23 ^b^	6.25 ± 0.21 ^b^

Different superscript letters indicate significant differences (^a,b^
*p* < 0.05). Control group, *n* = 10; 50 mg/kg DSF group, *n* = 10; 100 mg/kg DSF group, *n* = 10.

**Table 2 ijms-24-02371-t002:** Effects of DSF on mouse offspring.

Groups	Pregnancy Rate	Litter Size
Control	45% (9/20)	10.33 ± 1.73
DSF	67% (12/18)	13.55 ± 1.79

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
