# Peer review of "Effect of Disulfiram on the Reproductive Capacity of Female Mice"

_ijms, 2023, doi:10.3390/ijms24032371_

Round 1

Reviewer 1 Report

I like revised study. Idea is interesting and proposed methodology is convincing. I don't have any bigger doubts and concerns. I would suggest revision of the references because many of them, at least 24 out of 67 are 10 years old or more... Additionally, I suggest to go through the paper once again to fix some editorial mistakes, e.g. below every figure is "SEM, standard error of mean" while on the figures it is not used...

Author Response

Thank you for kind valuable suggestions. We have modified the references and some errors. Thank you very much. Wish you all the best in your life and work!

Reviewer 2 Report

The experiments described in the manuscript were well planned, thought out, and consistently characterized.

Please clarify:

1) since DSF is a drug with some side effects, have any side effects been observed during the experiments? If so, what are they?

2) why were the given doses of DSF, PMSG and hCG used?

3) please explain the abbreviations used in the manuscript, e.g. PMSG, hCG, ROS and etc.

4) I don't understand Figure 4; where are DSF doses marked? 

Author Response

(1)Since DSF is a drug with some side effects, have any side effects been observed during the experiments? If so, what are they?

--Thanks for your kind question. We did not find any obvious side effect on the female reproductive ability of mice in the test dose (50 mg / kg DSF). Besides, we observed that when the concentration of DSF exceeded 100 mg/kg, the ovulation rate and cleavage rate of fertilized oocytes in vitro began to decline. Although there was no statistically significant difference, the use of DSF exceeding the maximum safe dosage may have a negative effect on female reproductive capacity (L397-401).

(2)Why were the given doses of DSF, PMSG and hCG used?

--Thanks for your kind question. Based on the single maximum safe dose in humans, the mouse dose was calculated as 88 mg/kg (doi: 10.1016/j.cmet.2020.04.019). In this study, we used 25 mg / kg, 50 mg / kg and 100 mg / kg DSF for the test. The ovulation rate and cleavage rate of oocyte IVF in female mice began to decline when 100 mg / kg DSF was used. A significant increase in ovulation was observed in mice using 50 mg / kg DSF, so this concentration was chosen for testing in subsequent trials. In this study, we used PMSG combined with hCG to make mice to ovulate synchronously. A literature suggests that ovulation induction with 10 IU PMSG in combination with 10 IU hCG has no effect on embryo quality or embryo developmental potential in mice (doi: 10.1016 / s2095-3119 (13) 60325-1)

(3)Please explain the abbreviations used in the manuscript, e.g. PMSG, hCG, ROS and etc.

--Thank you very much for your kind valuable suggestions. We have already explained the abbreviations used in the manuscript.

(4)I don't understand Figure 4; where are DSF doses marked?

--Thanks for your kind question. We have changed the group names in Figure4 and annotated the DSF concentrations (L258).

Reviewer 3 Report

The manuscript has been reviewerd carefully. The topic is very interesting and of high importance in our nowadays society, in which postponing of first pregnancy has become normal.

L28: Please add the following reference: van der Reest J, Nardini Cecchino G, Haigis MC, Kordowitzki P. Mitochondria: Their relevance during oocyte ageing. Ageing Res Rev. 2021 Sep;70:101378. doi: 10.1016/j.arr.2021.101378.

L106: Please add the way of treatment application, either  per os, sub cut., i.m. , i.v. or i.p. ?

L136: Please add which stages of the oocytes, and how many of each treatment group  have been taken into consideration.

Figure 1 and 2: Please consider removing or transfering this figure as supplemental file. This is a standard procedure, and there is no need to show in the main MS since it does not reflect a crucial treatment outcome.

Figure 3: Please add the information at which day /night time the blood samples have been taken, since there could occur significant differences. Please explain if the blood samples have been taken only once or several times during a specific estrous phase since it is not stated in M&M.

Figure: It appears that the treatment rather positively works on the pituitary gland than directly on the follicle since E2 was lower compared to control groups. If FSH is increased there should be also higher rate of follicular development to antral follicles and in consequence higher E2. How do the authors explain this contradictionary result?

L266: Which cleavage stage the authors are referring to  and hwo was this rate counted? I assume the authors refer to cleavage rate of the zygotes not the oocytes?? Did they calculate cleavage rate from zygote to blastocyst stage, or how? Please fix, this is important to know!

L278: Again which stage of oocytes?

Figure 6A: The representative picture appears to me not in optimum focus and deepth. Please add a bright field picture to these fluorescence pictures. For this reviewer it appears as if the exposure time in the left picture was higher than in the right, therefore, very bright green fluorescence in the left. Please explain. Please explain also, how the relative fluores.inten. has been calculated or set, the mean of how many oocytes of field os view have been evaluated? It appears as if the control group was set as intensity valure "1" and the treatment group was compared to this control group. Please clarify.

Figure 7: please add bright field pictures. the right picture appears to have a more "blue" background color compared to the black background on the left, please explain why. Please describe how many oocytes have been evaluated and how the intesity was calculated.

L326: Please add the approx. number of cells which have been analyzed per/sample/donor.

General comment: The gene expression in granuulosa cells rather at a very low level reflect the quality of the oocyte and differs from the gene expression in oocyte. Please explain why the authors have not performed mRNA abundance in oocytes.

Main concern: For this reviewer it is not clear how the entire pool of collected oocytes have been distrubuted to the single approaches, meaning how many used for IVF, for ROS and so on, Please add this information to the figure legend respectively.  The sample size of 20 ovaries for each treatment group seems to be too low, taking into account the different experiments. Recommendation: Please add a larger number of oocytes!

Author Response

Dear Reviewer,

Thank you for your comments concerning our manuscript entitled “Effect of disulfiram on the reproductive capacity of female mice” (ID: ijms-2083825     ). Those comments are all valuable and very helpful for revising and improving our paper, as well as the important guiding significance to our research. We have studied the comments carefully and have made a correction which we hope meets with approval. Revised portions are marked in red on the paper. The main corrections in the paper and the responses to the reviewers and editor’s comments are as follows:

L28: Please add the following reference: van der Reest J, Nardini Cecchino G, Haigis MC, Kordowitzki P. Mitochondria: Their relevance during oocyte ageing. Ageing Res Rev. 2021 Sep;70:101378. doi: 10.1016/j.arr.2021.101378.

--Thanks for your kind suggestion. We have added this reference in L28.

L106: Please add the way of treatment application, either per os, sub cut., i.m. , i.v. or i.p. ?

--Thanks for your kind reminder. We have added the treatment way of mice in L102, L107, L118, L155 and L161.

L136: Please add which stages of the oocytes, and how many of each treatment group have been taken into consideration.

-- Thanks for your kind suggestion. We have added the period of oocytes and the information related to the number of oocytes used is in L134-135, L142-143, L261-262, L281, L290-291.

Figure 3: Please add the information at which day /night time the blood samples have been taken, since there could occur significant differences. Please explain if the blood samples have been taken only once or several times during a specific estrous phase since it is not stated in M&M.

-- Thank you very much for your suggestion. We selected those mice that were in diestrus phase for the test. After administration of DSF or corresponding solvent, we began to detect the estrous stages in mice at 8:00 am. When the estrous cycle of mice progressed to the next estrous phase, we collected the serum immediately and once for each mouse.

Figure: It appears that the treatment rather positively works on the pituitary gland than directly on the follicle since E2 was lower compared to control groups. If FSH is increased there should be also higher rate of follicular development to antral follicles and in consequence higher E2. How do the authors explain this contradictionary result?

--Thank you very much for your question. We also noticed that this result seemed contradictory. We are still thinking and studying about this phenomenon, hoping that it will be further explained in the subsequent studies.

L266: Which cleavage stage the authors are referring to and how was this rate counted? I assume the authors refer to cleavage rate of the zygotes not the oocytes?? Did they calculate cleavage rate from zygote to blastocyst stage, or how? Please fix, this is important to know!

-- Thank you very much for your question. Yes, we calculated the cleavage rate as the proportion of cleaved oocytes to 2-cell embryos after fertilization of oocytes that were at the MII stage. We consider that 2-cell embryos are already of value for embryo transfer. In future work, we will further investigate whether the efficacy and success rate of embryo transfer is improved after the use of DSF.

Figure 6A: For this reviewer it appears as if the exposure time in the left picture was higher than in the right, therefore, very bright green fluorescence in the left. Please explain. Please explain also, how the relative fluores.inten. has been calculated or set, the mean of how many oocytes of field os view have been evaluated? It appears as if the control group was set as intensity valure "1" and the treatment group was compared to this control group. Please clarify.

--Thank you very much for your question. Our exposure times were consistent. In the control group there were some oocytes with very high fluorescence intensity. It is common, because some oocytes present higher ROS levels due to the in vitro environment which is hyperoxic for them. Meanwhile, in the DSF group, there were fewer oocytes with this high level of reactive oxygen species, and the reactive oxygen species level of oocytes was more homogeneous. In both Figure 6 and Figure 7 trials, we used more than 30 oocytes per group, repeating 3 times(L142-143). We calculated the fluorescence intensity of each oocyte using image J software and figured out the mean for each group, while subtracting the calculated value of the oocyte free area in the background as the final statistic. The fluorescence intensity of the control group was set as 1.

Figure 7: please add bright field pictures. the right picture appears to have a more "blue" background color compared to the black background on the left, please explain why. Please describe how many oocytes have been evaluated and how the intensity was calculated.

--Thank you very much for your question. Our fluorescence pictures were all taken at the same wavelength. At the time of photographing, we found that when the fluorescence intensity of the oocytes became higher, the background color also became brighter, which seemed difficult to avoid. We recorded the corresponding value of fluorescence intensity of the oocytes by identifying each oocyte using the oval in Image J, while selecting 3 or 4 fields without oocytes in the background were recorded. Then the corresponding value of the fluorescence intensity of background region was subtracted from that of the oocyte to eliminate the influence of differences in background color between the two groups on the final results.

L326: Please add the approx. number of cells which have been analyzed per/sample/donor.

--Thank you very much for your suggestion. In the trials using ovarian granulosa cells for mRNA expression measurement, there were approximately 2*10^6 cells per sample. (L186-187)

General comment: The gene expression in granulosa cells rather at a very low level reflect the quality of the oocyte and differs from the gene expression in oocyte. Please explain why the authors have not performed mRNA abundance in oocytes.

-- Thanks for your valuable comments. During oocyte development and ovulation, ovarian granulosa cells play a crucial role, providing nutrition and essential cytokines, etc., for oocyte development and maturation. VEGF mRNA is mainly expressed in ovarian granulosa cells and theca cells (Kamat BR, Brown LF, Manseau EJ, Senger DR, Dvorak HF. Expression of vascular permeability factor/vascular endothelial growth factor by human granulosa and theca lutein cells. Role in corpus luteum development. Am J Pathol. 1995;146(1):157-165.). VEGF concentration is positively correlated with increased perifollicular vascular density, higher fertilization rate, better embryo quality and higher pregnancy rate (doi: 10.1007/s10815-008-9218-1.). Transcription ceases during the final stages of oocyte growth and resumes only when the embryonic genome is activated after fertilization (doi: 10.1126/science.abq4835.). Therefore, we used ovarian granulosa cells as a relatively ideal cell model to preliminarily explore the related molecular mechanisms by which DSF affects the reproductive ability of female mice. Your suggestions are of great significance, and we will further explore the effect of DSF on oocyte gene expression in the subsequent work to gain a more complete understanding of the molecular mechanisms underlying the effects of DSF on female reproduction.

Main concern: For this reviewer it is not clear how the entire pool of collected oocytes have been distrubuted to the single approaches, meaning how many used for IVF, for ROS and so on, Please add this information to the figure legend respectively. The sample size of 20 ovaries for each treatment group seems to be too low, taking into account the different experiments. Recommendation: Please add a larger number of oocytes!

-- Thank you for your valuable suggestions. We have annotated the number of mice or oocytes used for each trial in the figure legends. The ovaries of 20 mice per group were used only for ovulation tests. The mice and oocytes used for the other trials have been added in the figure notes of the respective trials as well as in the test methods. We will also replicate more experiments in our future work to make the data more accurate.

Finally, thank you for all your kind suggestions and questions. Wish you all the best!